# CO-EVOLVED SELF-CRITIQUE: ENHANCING LARGE LANGUAGE MODELS WITH SELF-GENERATED DATA

## ABSTRACT

Large language models (LLMs) have seen staggering progress in recent years. Contemporary LLMs rely on an immense amount of data for training, however, as LLMs continue to advance, the availability of high-quality external data is reaching a bottleneck, highlighting the need for model-generated data for further improvement. Although promising, directly utilizing the self-generated data for model training without scrutinized assessment or filtering can easily lead to deteriorated performance, or in other words, "garbage in, garbage out". In this study, our insight is to carefully craft a *self-critique* process, by equipping the LLMs with the ability to be self-aware and discriminative to the quality of its generated data. We introduce a co-evolved self-critique framework that enables an LLM to simultaneously enhance both its generative and evaluative capabilities through an iterative training process. This provides a scalable solution to ensure high-quality self-generated data and facilitate sustained model improvement. Fine-tuning Llama-3 models using this framework results in encouraging improvements in both instruction-following and discriminative abilities, demonstrating the effectiveness of our method.

## 1 INTRODUCTION

To date, the training of large language models (LLMs) has predominantly relied on external supervision data, such as human-annotated instructions and preferences, or those generated by advanced models, to perform Supervised Fine-Tuning (SFT) (Touvron et al., 2023; Dubey et al., 2024) and Reinforcement Learning from Human Feedback (RLHF) (Ouyang et al., 2022; Schulman et al., 2017). These approaches have been instrumental in shaping state-of-the-art LLMs, allowing them to achieve remarkable levels of performance across a wide range of tasks. However, as LLMs continue to advance, a critical challenge has emerged: the availability of external high-quality training data is becoming increasingly scarce, posing a bottleneck to the further scaling and improvement of strong AI models. As a result, there is a growing consensus that model training needs to transition toward using *self-generated* or *synthetic* data for training in order to sustain long-term performance growth in a bootstrapping manner.(Huang et al., 2022; Wang et al., 2022; Amodei et al., 2016; Burns et al., 2023; Zheng et al., 2024). As models become advanced and stronger, their self-generated data are expected to provide a viable source to supplement the limitations of external supervision data.

However, while self-generated data presents a promising path for the continued development of LLMs, it also poses significant challenges in maintaining the high quality of such data. Low-quality self-generated data, particularly data that lacks relevance or correctness, can severely degrade the model's performance, also known as "garbage in, garbage out" (Shumailov et al., 2024; Li et al., 2023a; Nakamoto et al., 2023; Valmeekam et al., 2023; Xu et al., 2022). Avoiding this pitfall requires the ability to discern which self-generated data is of high quality and which is not. Relying on external verifiers, reward models, or human feedback is becoming less effective as models grow to be more advanced, as these external verifiers often struggle to keep pace with the growing complexity and nuance of modern LLMs. One potential solution lies in the idea of *self-critique*, where the LLM itself plays an active role in evaluating the quality of its own outputs (Xu et al., 2024; Tian et al., 2024). Ideally, the model would become "self-aware" enough to recognize when its self-generated data is flawed or inadequate. This self-critique mechanism would allow advanced models to continually improve by autonomously filtering high-quality self-generated data, potentially overcoming the limitations of external validators.

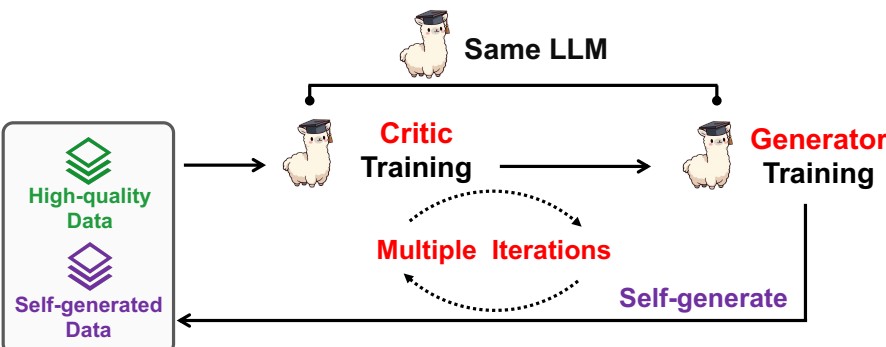

Figure 1: Overview of the co-evolved self-critique framework: both the generator and critic are iteratively trained using supervision signals within the co-evolvable training scheme.

Despite the conceptual appeal of self-critique, current approaches face several key drawbacks. These methods either treat the critique process as static (Yuan et al., 2024b; Sun et al., 2024; Li et al., 2023b; Guo et al., 2024) or overly trust the model's inherent discriminative capacity without explicit supervision (Wu et al., 2024). They focus on unilaterally improving LLM as *generator* (producing outputs for queries) while neglecting the explicit development of LLM as *critic* (evaluating the generator's outputs) in a reliable manner. Ideally, we want both the generator and the critic to co-evolve, improving each other through continuous interaction. This concept mirrors the co-evolution seen in Generative Adversarial Networks (GANs) (Goodfellow et al., 2020), where a stronger discriminator compels the generator to produce better outputs. In GANs, the feedback loop between the generator and discriminator is critical to its success, fostering substantial improvements beyond supervised learning and resulting in high-quality model generations. Applying this co-evolutionary notion to self-critique on LLMs could potentially allow models to simultaneously improve their generative and evaluative abilities, thus achieving greater performance gain. However, such a co-evolvable training scheme has not yet been fully explored in the current literature.

In this paper, we introduce *CoEvol*, a co-evolved self-critique framework designed to enable the same LLM to simultaneously serve as the generator and critic, and establish mutually evolvable improvement between these two sides (illustrated in Figure 1). Specifically, we hide a discriminative function (judging if a sample data is model generated or not) inside the LLM by reformulating it as a language task. At each iteration, using a small set of high-quality seed data along with newly self-generated data, we train the LLM to solve the discriminating task, thereby enhancing the capacity of the critic. Next, using a large volume of self-generated data, the critic filters data that is likely of high quality, and we then perform SFT on the same LLM, this time as the generator. Through this iterative procedure, both the generator and critic are explicitly and continuously trained on scrutinized supervision signals, enabling the co-evolution of the critic and generator.

In our experiments, we begin with the Llama-3-8B seed model and fine-tune it using the Ultra-Chat200k dataset (Ding et al., 2023). By applying the proposed co-evolved self-critique framework, the fine-tuned model demonstrates superior improvements over SFT and other related baselines in both instruction-following and discriminative abilities. Additionally, we empirically show the importance of the co-evolved training design and its advantages over prior self-improvement works (Yuan et al., 2024b), which underscores a promising direction for advancing LLMs with self-generated data.

## 2 RELATED WORK

**LLM Alignment** aims to ensure that increasingly powerful LLMs adhere to human values and intentions, preventing them from going out of control (Ji et al., 2023; Shen et al., 2023). Most of the studies focus on leveraging manual annotations or human feedback using techniques such as SFT or RLHF to achieve this object (Ouyang et al., 2022; Ji et al., 2023; Wang et al., 2024; Rafailov et al., 2024; Yuan et al., 2024a). As models approach the performance level of top human experts, obtaining high-quality annotations becomes prohibitively expensive, diminishing the effectiveness

of traditional methods. To address the challenge of maintaining effective oversight at such levels, there is a growing consensus that model training must transition toward using *self-generated* or *synthetic* data for scalable oversight, enabling long-term performance growth in a bootstrapping manner (Bai et al., 2022; Huang et al., 2022; Wang et al., 2022; Amodei et al., 2016; Burns et al., 2023; Zheng et al., 2024; Christiano et al., 2018).

**Self-generated** methods focus on utilizing data synthesized by the LLM itself to further enhance its performance. However, directly using data generated by the LLM itself may suffer from low quality, potentially leading to model collapse after training on such self-generated (Gerstgrasser et al., 2024; Li et al., 2023a; Nakamoto et al., 2023). Thus, Several approaches are raised to filter out low-quality data and can be mainly summarized into two categories: external filtering and self-critique. External filtering includes the use of external verifiers, heuristic rules, or external reward models (Wang et al., 2022; Li et al., 2023a; Valmeekam et al., 2023; Xu et al., 2022). However, as the generative capabilities of the LLM improve, these methods struggle to effectively assess the quality of data generated by the LLM itself, leading to potential failures in data filtering, making it necessary to explore self-critique methods that leverage the LLM itself for data quality evaluation.

**Self-critique** involves having the LLM evaluate its own output, helping it understand what constitutes a good response, thereby enhancing its own performance (Yuan et al., 2024b). Current self-critique methods can be categorized into static and dynamic two types: 1) The static self-critique method focuses on directly using the LLM itself to evaluate its own outputs. Specifically, some work focuses on directly informing the LLM of the general principles of good responses through prompts and allowing it to make judgments (Sun et al., 2024; Li et al., 2023b; Guo et al., 2024). Furthermore, Self-Rewarding utilizes LLM itself to evaluate its own responses, construct rewards, and then perform preference optimization (Yuan et al., 2024b). However, these efforts do not enhance the LLM's discriminative abilities alongside its generative capabilities, making it difficult for the LLM's own critic to assess quality as generative ability advances, thereby hindering further performance improvement. 2) The dynamic self-critique method aims to simultaneously enhance a model's instruction-following and discriminative capabilities. However, this promising approach has seen limited exploration. Existing works, such as Meta-rewarding (Wu et al., 2024), focus on using the LLM's judgment of its own evaluations to further refine its judgment accuracy. Unfortunately, this often leads to unreliable evaluations, as the secondary judgment can also be flawed, lacking any reliable supervision signal to effectively train the critic.

## 3 THE CO-EVOLVED SELF-CRITIQUE FRAMEWORK

In this section, we provide a detailed overview of the proposed CoEvol framework for LLMs. Similar to many existing approaches (Yuan et al., 2024b), we start with access to a base pretrained LLM, denoted as $M_0$, and a small set of high-quality seed dataset $\mathcal{D} = \{(x_i, y_i)\}_{i=1}^{D}$ (e.g., human expert-annotated data) for fine-tuning.

Unlike previous approaches (Yuan et al., 2024b; Wu et al., 2024), which only focus on improving the LLM as a *generator* (producing outputs for queries) rather than enhancing the LLM as a *critic* (evaluating the generator's outputs) in a reliable manner, our proposed framework aims to make both the generator and critic co-evolve. We provide explicit supervision signals to enhance the critic's discriminative capacity while training the generator and critic iteratively. Ultimately, this process yields a more powerful generator, trained on high-quality self-generated data. For clarity, starting with the base pretrained model $M_0$, we refer to the LLM as $M_g$ when it acts as the generator, and as $M_c$ when it acts as the critic.

### 3.1 CRITIC TRAINING

**Hide the discriminative function into the LLMs.** At each iteration, we have access to new self-generated data $\{(x_i, \hat{y}_i)\}$ produced by the current generator $M_g$, based on prompts (queries) $x_i$ sampled from the seed dataset $\mathcal{D}$. We are supposed to have an explicit discriminative function $\hat{l}$ that outputs $d \in (0, 1)$, which helps effectively differentiate between high-quality seed data and self-generated samples. The value of $d$ can be interpreted as a confidence level, indicating how closely a given sample aligns with high-quality seed data or self-generated content. This discriminative function is typically optimized using cross-entropy loss to solve the discrimination task:

$$\mathcal{L} = -\sum_i \left[ l(\tilde{x}_i, \tilde{y}_i) \log(\hat{l}(\tilde{x}_i, \tilde{y}_i)) + (1 - l(\tilde{x}_i, \tilde{y}_i)) \log(1 - \hat{l}(\tilde{x}_i, \tilde{y}_i)) \right] \tag{1}$$

where $l(\tilde{x}_i, \tilde{y}_i)$ represents the true label of the sample $(\tilde{x}_i, \tilde{y}_i)$, while $\hat{l}(\tilde{x}_i, \tilde{y}_i)$ is the predicted label. Specifically, when $(\tilde{x}_i, \tilde{y}_i) \sim \{(x_i, y_i)\}_{i=1}^{D}$, meaning the sample comes from the seed dataset, $l(\tilde{x}_i, \tilde{y}_i) = 1$; and when $(\tilde{x}_i, \tilde{y}_i) \sim \{(x_i, \hat{y}_i)\}$, meaning the sample is self-generated, $l(\tilde{x}_i, \tilde{y}_i) = 0$.

We aim for the LLM, acting as the critic $M_c$, to possess this discriminative capability, meaning the discriminative function should be hidden into $M_c$. However, the aforementioned loss cannot be directly optimized within the LLM framework. To address this, we reformulate the original discriminative task as a language task, constructing a training dataset designed for $M_c$ to perform Supervised Fine-Tuning (SFT).

We refer to this dataset for critic training as $\mathcal{D}_c$. For each sample in the sets $\{(x_i, y_i)\}$ and $\{(x_i, \hat{y}_i)\}$ at each iteration, we perform the following steps: (i) format the samples into the judge prompt template (see Figure 2) to create the judge prompt $j_i^c$; (ii) assign the identifiers (labels) "M" or "m" as $l_i^c$, depending on whether the output (response) of the sample is $y_i$ or $\hat{y}_i$, respectively. This process allows us to construct the training dataset as $\mathcal{D}_c = \{(j_i^c, l_i^c)\}$.

---

**Judge prompt $j_i^c$**

Review the user's query and the corresponding response.

User: {prompt}

Response: {response}

After evaluating the quality and relevance of the response, determine whether it was generated by a human expert (identifier: M) or by yourself (identifier: m). Your output should consist of only one of these identifiers: M or m.

---

**Response $l_i^c$**

M or m

---

Figure 2: The template of judge prompt $j_i^c$ and response $l_i^c$.

After constructing the dataset $\mathcal{D}_c$, we perform SFT on the critic $M_c$ using this dataset, resulting in an updated version of the LLM in its critic role. After critic training, given a judge prompt $j_i^c$, the trained critic $M_c$ outputs a token, which corresponds to the labels "M" or "m". The label "M" indicates that $M_c$ believes the sample is high-quality seed data, likely generated by a human expert, while "m" suggests the sample was generated by the current generator $M_g$. By providing real-time and explicit supervision signals (through seed data and newly self-generated data), the trained critic offers a more reliable evaluation of the newly self-generated data compared to previous approaches that treat the process as static or rely solely on the LLM's inherent capabilities without explicit critic training.

At this point, we have not yet obtained the confidence level $d$. We follow these steps to compute $d$: for a given judge prompt $j_i^c$, we send it to the critic and examine its logits output. We then extract the logits corresponding to the tokens "M" and "m" from the output. Applying the softmax function to these logits, we interpret the resulting value for "M" as the confidence level $d \in (0, 1)$.

## 3.2 GENERATOR TRAINING

Once the critic has been trained to accurately discriminate the quality of the current self-generated data, we can leverage this capability to enhance the generator $M_g$. Within the CoEvol (co-evolved self-critique) framework, there are several possible methods for achieving this. Here, we propose

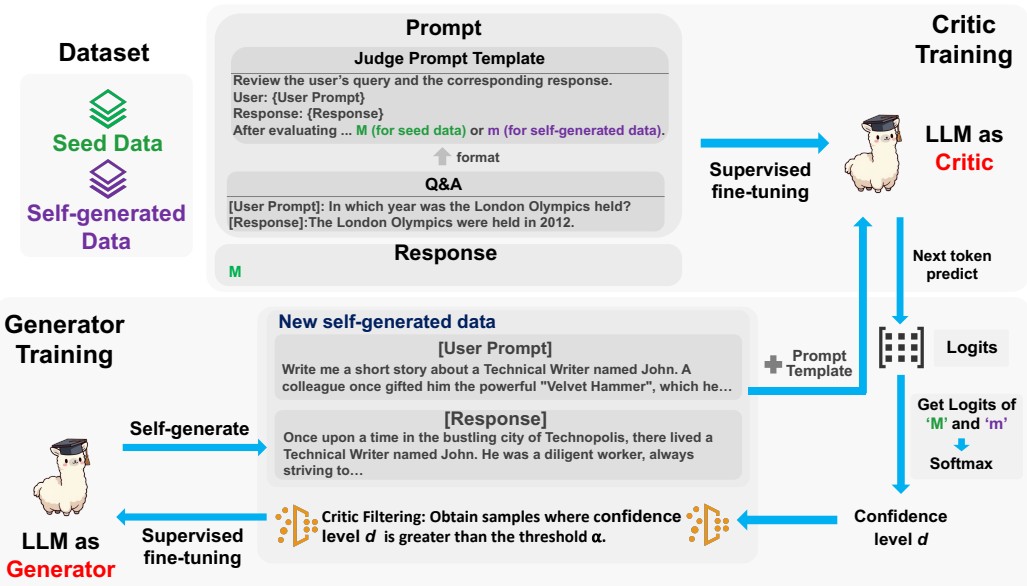

Figure 3: The specific co-evolved self-critique training process: at each iteration, the high-quality seed data, combined with newly self-generated data, are used to train the LLM as a critic. Then, in the same iteration, the LLM as a generator is trained using the self-generated data filtered by the critic.

a simple yet effective approach: using the confidence level $d$ provided by the critic $M_c$ to filter the self-generated data, and then applying Supervised Fine-Tuning (SFT) on $M_g$ using the filtered data.

**Critic Filtering.** Specifically, for each new self-generated sample $(x_i, \hat{y}_i)$ produced by $M_g$, we format it into the judge prompt template to create the judge prompt $j_i^c$. This prompt is then passed to the critic LLM to produce the confidence level $d_i$. If $d_i > \alpha$, where $\alpha$ is a hyperparameter controlling the selection threshold, the self-generated sample $(x_i, \hat{y}_i)$ is added to the augmented dataset. Finally, we perform SFT on the current LLM using this augmented dataset, thereby improving the $M_g$ iteratively based on high-confidence self-generated samples.

Starting from the base pretrained model $M_0$, we iteratively perform both the critic training and generator training, formulating a co-evolved self-critique training process. The overall training procedure is summarized in Algorithm 1. The specific training procedure at each iteration is illustrated in Figure 3.

---

**Algorithm 1** Co-evolved self-critique training

**Input:** The base pretrained model $M_0$ (act as both generator and critic), the high-quality human-annotated seed dataset $\mathcal{D}$.

**for** $t = 0, 1, 2, ..., T$ **do**

    // Critic training

    Sample $\{(x_i, y_i)\}_{i=1}^{\mathcal{B}^c} \sim \mathcal{D}$. Generate $\{(x_i, \hat{y}_i)\}_{i=1}^{\mathcal{B}^c}$.

    Construct the dataset $\mathcal{D}^c = \{(x_i^c, y_i^c)\}_{i=1}^{2 \times \mathcal{B}^c}$.

    Update the LLM by performing supervised fine-tuning using $\mathcal{D}^c$.

    // Generator training

    Sample $\{(x_i, y_i)\}_{i=1}^{\mathcal{B}^g} \sim \mathcal{D}$. Generate $\{(x_i, \hat{y}_{ij}) \mid i = 1, \ldots, \mathcal{B}^c, j = 1, \ldots, N\}$.

    Perform critic filtering to construct $\mathcal{D}^g$.

    Update the LLM by performing supervised fine-tuning using $\mathcal{D}^g$.

**end for**

---

## 4 EXPERIMENTS

CoEvol provides both the generator and critic with explicit and continuous supervision signals. We design experiments to assess the effectiveness of CoEvol in improving both the generator and critic. For the generator, we evaluate its instruction-following ability, which refers to the LLM's capacity to generate high-quality, helpful, harmless, relevant, and clear responses to given prompts. For the critic, we evaluate its discriminative ability, which refers to the LLM's capacity to accurately determine whether a given sample is high-quality (generated by a human expert) or self-generated. Specifically, our experiments are designed to address the following questions:

- Given a limited amount of high-quality data, can CoEvol improve the instruction-following ability more effectively than traditional supervised fine-tuning (SFT) methods?
- Beyond instruction-following, can CoEvol progressively enhance discriminative ability compared to approaches that treat the critic process as static, as seen in other self-critique methods?

### 4.1 EXPERIMENTAL SETUP

**Base model & seed data.** In our experiments, we use Llama-3-8B as the base pretrained model $M_0$. For the high-quality seed dataset $\mathcal{D}$, we select 4,000 samples from the UltraChat200k dataset (Ding et al., 2023), with each sample containing a prompt-response pair.

**Training details.** We follow the training procedure outlined in Algorithm 1. In each training iteration, we sample $\mathcal{B}^c = \mathcal{B}^g = 1000$ examples from the seed dataset $\mathcal{D}$. During critic training, LLM generates a response $\hat{y}_i$ for each prompt using a temperature of 0.8 and top-p of 0.95. For generator training, LLM generates $N = 3$ response variations per prompt with the same temperature and top-p settings. In critic filtering, we apply a temperature $\tau = 2.5$ in the softmax function and set the threshold $\alpha = 0.55$. We begin with the base Llama-3-8B model and iteratively train it as both the generator and critic over the course of $T = 4$ iterations.

### 4.2 EVALUATION METRICS AND BASELINES

**Generator's Instruction Following Ability**.

Similar to (Li et al., 2023b), we build a test prompt dataset by sourcing prompts from TruthfulQA (Lin et al., 2021), ShareGPT (Chiang et al., 2023), Evol-Instruct (Xu et al., 2023), Open Assistant (Köpf et al., 2024), and additional prompts crowdsourced from the authors. This ensures broad coverage across diverse task categories, including writing, coding, mathematical reasoning, and safety.

We randomly select 256 prompts from this dataset to evaluate our proposed method against the baseline model. For the evaluation, we employ the LLM-as-a-Judge (Dubois et al., 2024; Li et al., 2023c; Zheng et al., 2023), using GPT-4 as the evaluator with the AlpacaEval evaluation prompt (Li et al., 2023c). The prompts are presented in both orders for pairwise comparison. If GPT-4's judgments conflict, we classify the result as a tie. For more details about the GPT-4 evaluator we used, please refer to Appendix A.1. Additionally, we conduct a similar assessment with human evaluators to further validate the results.

**Baseline.** For instruction following ability, the main baselines we compare to are:

- *SFT Baseline*: This baseline represents the model fine-tuned from the base pretrained Llama-3-8B using the seed dataset $\mathcal{D}$ via supervised fine-tuning (SFT). To ensure a fair comparison, we use the same seed data as CoEvol at each iteration for SFT, resulting in the corresponding SFT baseline model $M_t$, where $t$ denotes the iteration.
- *SFT Baseline* ++: This baseline represents the model fine-tuned from the base pretrained Llama-3-8B using a larger dataset via SFT. In addition to the 4,000 samples from the seed dataset, we include an additional 6,000 samples from the UltraChat200k dataset, resulting in a total of 10,000 samples. Since the model is fine-tuned on a larger dataset, it is expected to perform better than the SFT Baseline, hence we label this as SFT Baseline ++.

**Critic's discriminating Ability**.

To evaluate whether the critic can accurately determine if a given sample is high-quality (generated by a human expert) or self-generated, we randomly select 500 samples from the UltraChat200k dataset as the critic test data. After each iteration of critic training, we generate new responses from the LLM for the prompts in the critic test data. These data are then formatted into a judge prompt template and sent to the critic, which provides the predicted label. We measure the classification accuracy based on these predictions.

**Baseline.** For discriminating ability, the main baseline we compare to is:

- *Self-rewarding*: This baseline represents methods such as self-rewarding (Yuan et al., 2024b), where the LLM is initially trained as a critic, and its discriminative ability is used directly without further critic training. Subsequent training focuses solely on improving the instruction-following ability.

## 4.3 GENERATOR'S INSTRUCTION FOLLOWING EVALUATION

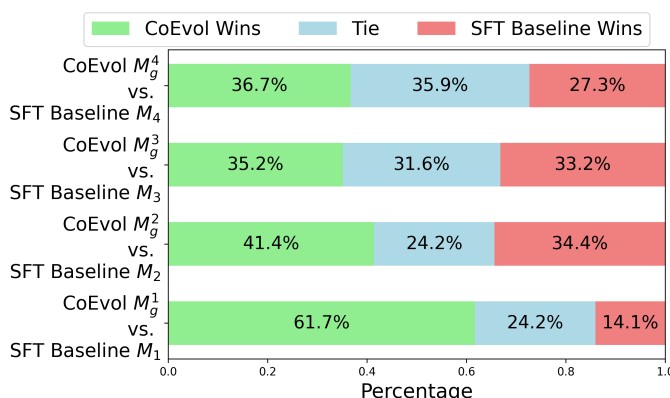

Figure 4: The comparison of instruction following ability between CoEvol model and SFT Baseline, evaluated via LLM-as-a-Judge. CoEvol outperforms SFT Baseline across iterative model versions, showing consistently gains. CoEvol $M_g^n$ refers to the generator at the n-th iteration, while SFT Baseline $M_n$ represents the model trained via SFT using the same seed data at the n-th iteration.

**Comparison to SFT Baseline.** The comparison results with the SFT baseline, using LLM-as-a-Judge, are presented in Figure 4. The results demonstrate that the CoEvol model consistently outperforms the SFT Baseline across all iterations, with the most substantial improvement seen in the first iteration, where CoEvol wins 61.7% of the comparisons. As the iterations progress, CoEvol continues to show an advantage. These results indicate that the CoEvol framework maintains consistent gains over the SFT Baseline across different model versions. We also provide examples of self-generated sample by the generator over the four iterations in Appendix A.2.

**Comparison between CoEvol models from adjacent iterations.** The comparison results of instruction-following ability between successive iterations of the CoEvol model, are presented in Figure 5. The results show a consistent trend of improvement in newer model versions ($M_g^{n+1}$) over their previous counterparts ($M_g^n$).

**Comparison to SFT Baseline ++.** Our CoEvol model can even outperform models fine-tuned on larger datasets (Figure 6). Despite the data advantage of the SFT Baseline ++, the CoEvol models demonstrate competitive performance, eventually surpassing the Baseline by the fourth iteration. In the final comparison ($M_g^4$ vs. SFT Baseline ++), CoEvol wins 26.6% of the time compared to 17.6% for the SFT Baseline++, showing CoEvol's capacity to achieve superior results with less data.

### 4.3.1 HUMAN EVALUATION

To further demonstrate the effectiveness of our proposed CoEvol, we conducted a human evaluation comparing the results generated by CoEvol with those produced by the SFT baseline across all iterations just as demonstrated in Figure 7. In detail, we randomly sample 20 examples from 256

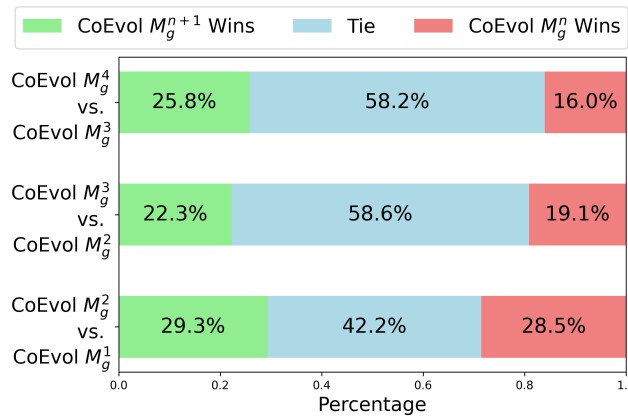

Figure 5: The comparison of instruction-following ability between CoEvol models from adjacent iterations, evaluated via LLM-as-a-Judge, demonstrates consistent improvement in instruction-following ability.

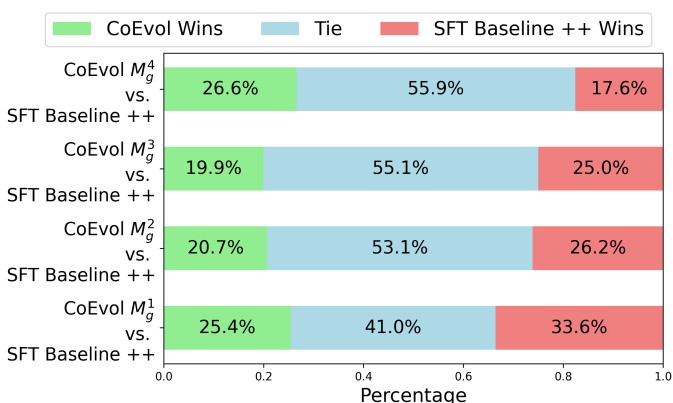

Figure 6: The comparison of instruction-following ability between CoEvol models and the SFT Baseline ++. The results show that, despite the SFT Baseline ++ being trained with 2.5 times more data than CoEvol, our CoEvol model outperforms it by the fourth iteration.

test data for human evaluation. The university-educated annotator with a bachelor's degree is tasked with labeling the data, determining which model's output is better or if the two models are tied.

The results in Figure 7 demonstrate that our CoEvol method consistently achieves higher win rates compared to the SFT baseline in each iteration. This outcome aligns with the results obtained from evaluations conducted using LLM-as-a-Judge, further proving the effectiveness of our proposed CoEvol method. We can also observe that the overall trend of human evaluation aligns with the results from LLM-as-Judge. Although the proportion of ties in human evaluation is slightly higher than in LLM-as-Judge, this outcome may be attributed to the model's generation capabilities being sufficiently advanced. As a result, it becomes challenging for humans to discern which model's output is better or worse, making this a reasonable outcome.

## 4.4 CRITIC'S DISCRIMINATING EVALUATION

We evaluate the critic's discriminative ability using the critic test data. At each iteration, the LLM generates new self-produced data, and as the LLM's instruction-following ability improves, its self-generated outputs may become more similar to the high-quality seed data. The classification accuracy in determining whether a given critic test sample is high-quality (generated by expert human) or self-generated across four iterations is shown in Figure 8. The CoEvol method maintains consistently high accuracy, starting at 83.3% and stabilizing around 69% in later iterations. In contrast, the

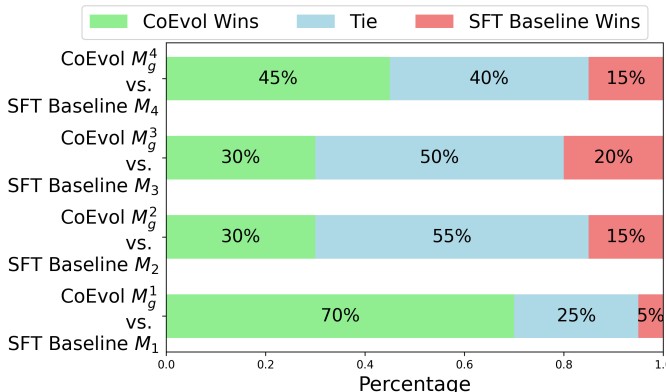

Figure 7: The comparison of instruction-following ability between our proposed CoEvol and SFT baseline under human evaluation. The experimental results are similar to those of LLM-as-a-Judge, demonstrating that our CoEvol achieves better performance compared to the SFT baseline.

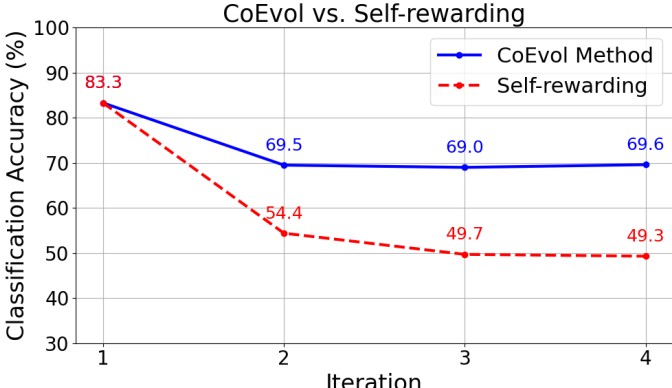

Figure 8: The comparison of discriminative ability between CoEvol and self-rewarding method shows that, as iterations increase and the model's instruction-following ability improves, the classification accuracy of the self-rewarding method significantly drops. This underscores the importance of continuous and iterative training of the critic.

self-rewarding method shows a sharp decline, dropping from 54.4% in the first iteration to 49.3% by the fourth iteration. These results underscore the superior performance of CoEvol and highlight the importance of continuous, iterative training for improving discriminative ability.

We also observe a decrease in classification accuracy for both methods at the second iteration. This may be due to the improvement in the model's instruction-following ability, which makes it more challenging to distinguish between high-quality seed data and self-generated data. However, CoEvol shows a significantly smaller drop in accuracy compared to the self-rewarding method, where the discriminative ability is used directly without further critic training. With CoEvol, we can more effectively select appropriate self-generated samples for the next generator training, maintaining better overall performance.

## 5 CONCLUSION

In this paper, we introduced a novel co-evolved self-critique framework, namely CoEvol, that allows LLMs to simultaneously enhance their generative and evaluative capacities through multiple iterations. Our results demonstrate that CoEvol consistently outperforms traditional SFT baselines, even when the baseline is trained with significantly more data. The comparative results between CoEvol

models and self-rewarding methods further underscore the importance of continuous critic training, as self-rewarding models suffer from a notable decline in classification accuracy over time.

There are also some limitations to this approach. One of the primary challenges is the increased computational cost of training both the generator and critic in each iteration. This dual training process requires more resources and time, which may limit its scalability for extremely large models or datasets. Future work should focus on addressing these limitations by refining the co-evolution process and exploring ways to reduce computational overhead while maintaining or enhancing performance. Additionally, we plan to apply our method to larger base pretrained models to demonstrate the potential for scalable oversight and further improve model capabilities.

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

# A APPENDIX

## A.1 EVALUATING DETAILS

Specifically, we use the *alpaca_eval_gpt4_turbo_fn* evaluator from AlpacaEval (Li et al., 2023c) for LLM-as-Judge. Figure 9 shows the judging prompt. Note that it differs from the judge prompt designed for our CoEvol framework.

---

**Judging prompt of alpaca_eval_gpt4_turbo_fn**

```
<im_start>system
```
You are a highly efficient assistant, who evaluates and rank large language models (LLMs) based on the quality of their responses to given prompts. This process will create a leaderboard reflecting the most accurate and human-preferred answers.
```
<im_end>
<im_start>user
```
I require a leaderboard for various large language models. I'll provide you with prompts given to these models and their corresponding responses. Your task is to assess these responses, ranking the models in order of preference from a human perspective. Once ranked, please output the results in a structured JSON format for the make_partial_leaderboard function.

## Prompt

```
{
    "instruction": """{instruction}""",
}
```
## Model Outputs

Here are the unordered outputs from the models. Each output is associated with a specific model, identified by a unique model identifier.

```
{
    {
        "model": "m",
        "output": """{output_1}"""
    },
    {
        "model": "M",
        "output": """{output_2}"""
    }
}
```
## Task

Evaluate and rank the models based on the quality and relevance of their outputs. The ranking should be such that the model with the highest quality output is ranked first.
```
<im_end>
```

---

Figure 9: The Judging prompt of alpaca eval_gpt4_turbo_fn.

## A.2 GENERATION SAMPLES

### A.2.1 SELF-GENERATED DATA FROM COEVOL

We provide examples of selected self-generated data from CoEvol at four iterations.

**Self-generated data example at iteration 1.**

Prompt:

Can you provide a list of the file formats that the program supports for exporting reports?

Generate according to: several file formats. Another command helps you to walk through the alignment and create an animation.

The program computes volumes of under and over-excavation by comparing project and measured cross-sections. Reports can be generated and printed or exported to Excel and other file formats.

Every project is compound of an horizontal and vertical alignment, Super elevation, templates and point files, supporting different formats.

The Powerful cross-section editor has tools for erasing, adding or moving points before drawing.

---

Response:

Hello, <User name>! The following formats are supported for exporting reports: PDF, Word, Excel, Image, Text, HTML, RTF, and CSV.

The animation tool allows you to view the project animation and walk through the alignment in order to visualize the changes in the design and measure cross-sections at any point along the alignment.

The program generates reports that show the volumes of under and over-excavation by comparing the project and measured cross-sections. You can print the reports or export them to Excel or other file formats.

Each project consists of a horizontal and vertical alignment, super elevation, templates, and point files that support different formats.

The cross-section editor has tools for erasing, adding, or moving points before drawing the cross-sections.

If you need any further assistance, please let me know. I'm here to help you.

A.2.2    A MISJUDGED SAMPLE FROM THE SELF-REWARDING METHOD

This is a misjudged sample from the self-rewarding method at the fourth iteration of CoEvol. Although the response is self-generated, the self-rewarding method incorrectly judged it as being produced by a human.

756
757
758
759
760
761
762
763
764
765
766
767
768
769
770
771
772
773
774
775
776
777
778
779
780
781
782
783
784
785
786
787
788
789
790
791
792
793
794
795
796
797
798
799
800
801
802
803
804
805
806
807
808
809

---

**Self-generated data example at iteration 2.**

Prompt:

Explain the traditional use of sunflower seeds in Native American societies, and how it has impacted modern agriculture.

---

Response:

Sunflower seeds have played a significant role in Native American societies for centuries. The seeds were traditionally used for a variety of purposes, including food, medicine, and oil production.

For food, sunflower seeds were often roasted or ground into flour to make bread or porridge. They were also eaten raw or mixed with other ingredients to make a range of dishes. In some Native American cultures, sunflower seeds were considered a symbol of peace and were used as a ceremonial offering.

Sunflower seeds were also used for medicinal purposes. They were believed to have healing properties, and were used to treat a variety of ailments, including respiratory problems, skin conditions, and digestive issues. In some cultures, sunflower seeds were also believed to have aphrodisiac properties.

In addition to their use in food and medicine, sunflower seeds were also used for oil production. The oil extracted from sunflower seeds was used for cooking, as well as for lighting and lubrication. The oil was also used to treat leather and make soap.

The traditional use of sunflower seeds in Native American societies has had a significant impact on modern agriculture. Today, sunflower seeds are grown commercially for their oil, which is used in a variety of products, including cooking oil, margarine, and biodiesel. Sunflower seeds are also used as a source of protein and as a snack food.

---

810
811
812
813
814
815
816
817
818
819
820
821
822
823
824
825
826
827
828
829
830
831
832
833
834
835
836
837
838
839
840
841
842
843
844
845
846
847
848
849
850
851
852
853
854
855
856
857
858
859
860
861
862
863

**Self-generated data example at iteration 3.**

Prompt:

Write a book report that summarizes and analyzes the themes and writing style of a memoir that explores the writer's experience living with chronic illness. In your report, consider how the writer's personal story can shed light on larger issues surrounding chronic illness, such as access to healthcare, stigma, and disability rights. You may also want to discuss how the author's use of language, imagery, and narrative structure contribute to their overall message about living with a chronic condition. Your report should be well-organized, persuasive, and demonstrate a strong understanding of the text.

Response:

Title: "Living with Chronic Illness: A Memoir"

Summary: This memoir tells the story of a woman who has lived with a chronic illness for many years. The author describes the challenges she has faced in her daily life, from managing her symptoms to navigating the healthcare system. She also discusses the stigma she has faced from others who don't understand her condition, and how this has affected her self-esteem and sense of identity.

Themes: The main theme of this memoir is resilience. The author describes how she has learned to cope with her illness, despite the many obstacles she has faced. She also discusses how her illness has made her more aware of the inequalities in our healthcare system, and how this has motivated her to advocate for better access to care for all people living with chronic illnesses.

Writing Style: The author's writing style is engaging and personal. She uses vivid imagery and metaphor to describe her experiences, making the reader feel as if they are right there with her. She also uses humor to lighten the mood and make her story more accessible to readers who may not have personal experience with chronic illness.

Analysis: This memoir is a powerful testament to the challenges and triumphs of living with a chronic illness. The author's honesty and vulnerability make her story relatable to anyone who has ever faced a difficult challenge. Her message about resilience and advocacy is inspiring, and her insights into the larger issues surrounding chronic illness are thought-provoking and important. Overall, this memoir is a must-read for anyone who wants to understand the experience of living with a chronic condition.

**Self-generated data example at iteration 4.**

Prompt:

Using Swift, create a language translation app that enables users to type in text in their chosen language and display the translated text in another language of their choice. The app must support multiple languages, and the translation should be accurate and reliable. The interface must be user-friendly with clear instructions, and the app must be able to save translation history for future reference. Additionally, the app should have an option for voice input and output for the convenience of the user.

Response:

Here is a possible implementation of a language translation app in Swift:

1. Create a list of supported languages with their corresponding API keys for translation.

2. Use the 'TextField' to allow users to enter text in their chosen language.

3. Use the 'Button' to start the translation process.

4. When the button is pressed, the app should make a request to the selected language's API to translate the text.

5. The translated text should be displayed in a 'Label' or 'TextView'.

6. Add an option to save the translation history, so users can access previous translations.

7. Include a voice inputoutput option, so users can translate text using their voice instead of typing.

8. Add a settings page where users can choose their preferred language, set up a translation history, and enabledisable voice inputoutput.

9. Test the app thoroughly to ensure accurate and reliable translations, and a user-friendly interface.

10. Implement the app using best practices, such as using 'structs' and 'enums' to store data, and using 'guard' statements to handle errors and edge cases.

918
919
920
921
922
923
924
925
926
927
928
929
930
931
932
933
934
935
936
937
938
939
940
941
942
943
944
945
946
947
948
949
950
951
952
953
954
955
956
957
958
959
960
961
962
963
964
965
966
967
968
969
970
971

---

**A misjudged sample from the self-rewarding method.**

**Prompt:**

**Review the user's query and the corresponding response.**
**User:**
Write a comprehensive and step-by-step guide on becoming a successful virtual bookkeeper, including details on skills required, software and tools to use, strategies for marketing and growing your business, tips for managing clients, and examples of successful bookkeeping businesses in the virtual space. Ensure that the guide is well-structured, easy to follow, and provides practical advice for anyone looking to start their virtual bookkeeping career.
**Response:**
Here is a comprehensive guide on becoming a successful virtual bookkeeper:
**Skills Required**
To become a successful virtual bookkeeper, you will need to have strong mathematical and accounting skills, as well as excellent communication and organizational abilities. You should also be proficient in using accounting software and other bookkeeping tools.
**Software and Tools to Use**
There are several different types of software and tools available for virtual bookkeepers to use, including accounting software such as QuickBooks, bookkeeping apps like Wave, and online invoicing systems like Invoice2go. It's important to choose the right tools for your business and to ensure that they are compatible with your workflow and needs.
**Strategies for Marketing and Growing Your Business**
Marketing and growing your virtual bookkeeping business is essential for success. You can use a variety of strategies, such as building a website, creating social media profiles, networking with other professionals, and participating in online forums and communities. Additionally, you can offer discounts and promotions to attract new clients and build your reputation.
**Tips for Managing Clients**
Managing clients effectively is key to running a successful virtual bookkeeping business. Some tips include setting clear expectations, communicating regularly with clients, being responsive to their needs, and providing high-quality service. It's also important to maintain good relationships with your clients by being friendly and helpful, and offering personalized services whenever possible.
**Examples of Successful Bookkeeping Businesses in the Virtual Space**
There are many successful virtual bookkeeping businesses out there, including companies like Bookkeeper360, Invoicely, and Wave Accounting. These businesses offer a range of services, from basic bookkeeping to more complex financial management and advisory services. By studying these businesses and learning from their success, you can develop your own unique approach to virtual bookkeeping and create a successful business of your own.
I hope this guide has been helpful in providing you with a comprehensive overview of what it takes to become a successful virtual bookkeeper. Remember, success requires hard work, dedication, and a commitment to providing high-quality service to your clients. With the right skills, tools, and strategies, you can build a successful virtual bookkeeping business and make a positive impact on the lives of your clients and the industry as a whole.
**After evaluating the quality and relevance of the response, determine whether it was generated by a more advanced model (identifier: M) or by yourself (identifier: m). Your output should consist of only one of these identifiers: M or m.**

---

**Response:** M.

