# OpenReview forum: "Co-evolved Self-Critique: Enhancing Large Language Models with Self-Generated Data"
_ICLR.cc/2025/Conference — ICLR 2025 Conference Withdrawn Submission_

### Official Review · Reviewer_ZVQq · 2024-10-17

**Soundness:** 2
**Presentation:** 3
**Contribution:** 2
**Rating:** 3
**Confidence:** 4

**Summary:**

This paper emphasizes the current neglect of optimizing LLM judging abilities in self-critique work and draws inspiration from the idea of GANs to iteratively train both the generate and judge capabilities of LLMs, proposing the CoEvol framework. The authors' experiments confirm that, in each round of iterative training under the CoEvol framework, Llama-3-8B outperforms the SFT method.

**Strengths:**

The core contribution of this paper lies in demonstrating that enhancing the discriminative ability of LLMs can improve the process of continuously optimizing model performance using the self-critique method. Additionally, the paper draws on the idea of GANs to train both the model's discriminative and generative abilities, which is an interesting approach.

**Weaknesses:**

However, this paper has the following issues:

1. When compared to the SFT method, although the seed data for training the generation ability was aligned, CoEvol strengthens learning from high-quality human responses through an additional process of training the critic. Therefore, the comparison is not entirely fair, as the data amount the model encounters in each round is not truly aligned.

2. Was the seed data included to construct dataset $D^g$? If not, the dataset for training the generation ability might not be as high-quality as human responses, so why does it perform better?

3. The experimental setups in the paper are not very convincing:
   - There is a lack of hyperparameter ablation, particularly noting the unusual values such as setting \(\alpha\) to 0.55 and the temperature to 2.5.
   - It is not stated (or I apologize if I missed it) how many times the experiment was conducted and whether the results are statistically significant.
   - Considering that Meta-rewarding also emphasizes improving the quality of self-generated rewards during the self-critique process, it is essential to include it as a baseline for comparison.

**Questions:**

See weakness above.

---

### Official Review · Reviewer_4GKR · 2024-10-28

**Soundness:** 2
**Presentation:** 2
**Contribution:** 3
**Rating:** 3
**Confidence:** 4

**Summary:**

The paper proposes a framework, CoEvol, for training on synthetic data. Unlike previous approaches, which first train a critic model and then utilize it to filter high-quality self-generated data for further training, the method proposes to iteratively train the critic as well, starting from a collection of seed data.

The approach outperforms the baselines on the tasks shown, both using LLM-as-a-judge and human evaluations.

A comparison analysis between the critic classification accuracy of the Self-Rewarding vs. CoEvol framework shows that the latter's performance deteriorates less rapidly as the model generations hypothetically become harder to distinguish from the ground-truth examples.

**Strengths:**

**Idea.** \
Good idea that can be potentially powerful. Iteratively improving the model as a critic in addition to as a generator makes sense. This is also an important area in current LLMs. It is important to understand the capabilities and limitations of models to self-improve using their own data, and devise better methods.

**Results** \
Some results were shown, particularly preference comparisons, following the Self-Rewarding paper format. Both LLM-as-a-judge and human evaluations are presented, and the approach outperforms the baselines by a reasonable amount.

**Weaknesses:**

**Experiments and baselines**

Only 1 model was used, a Llama-3-8b. While I understand the difficulty in training more and larger models, it is important to show that the results transfer to more models.

In Figure 5, performance gap seems to decrease instead of increase as the training iterations increase. This result goes against the intuition of the method that iterative refinement of both the critic and generator should improve performance.

Crucially, while in 4.4 Section a comparison was made with the Self-Rewarding method, no such baseline was introduced in the previous section. Not comparing to the key comparison method raises concerns about the validity of the method and study, especially since the whole premise is predicated on the difference between the added component of training the critic vs. not in the Self-Rewarding paper.

**Unclear Details** \
I was unclear on how the critic is trained on subsequent iterations, since it has already seen the expert data in the first iteration.

**Questions:**

Do you use the same seed human-expert data to compare with the self-generated data at each iteration? How does that work, since the model has already seen the human-expert data? Could you please explain this process in more detail?

---

### Official Review · Reviewer_u2Yj · 2024-10-31

**Soundness:** 3
**Presentation:** 3
**Contribution:** 3
**Rating:** 5
**Confidence:** 4

**Summary:**

The paper addresses a key challenge in LLM development: as models get stronger, high-quality external training data becomes scarce, making self-generated data increasingly important. However, using self-generated data risks "garbage in, garbage out" if not properly filtered.
The authors propose CoEvol, a framework where the same LLM acts as both generator and critic, with both roles improving through iterative training. The key innovation is treating the discriminative task (identifying high vs low quality responses) as a language task within the LLM. At each iteration:

The critic is trained to distinguish between high-quality seed data and self-generated data
The generator creates new responses
The critic filters these responses based on quality
The generator is trained on the filtered high-quality responses

**Strengths:**

The paper addresses an important challenge in LLM training - how to effectively use self-generated data as high-quality external data becomes scarce. This is particularly relevant as models continue to advance.

The framework's design is novel and theoretically appealing:
Combines generator and critic roles in the same LLM
Uses iterative co-evolution between these roles
Reformulates discriminative tasks as language tasks

**Weaknesses:**

1  Concerning Trends in Results:
The accuracy gap between CoEvol and SFT baseline narrows in later iterations (Fig 4), suggesting diminishing returns
The discriminative accuracy doesn't improve across iterations - in fact, it drops from 83.3% to around 69% (Fig 8)
The authors don't adequately explain or address these trends, which may indicate fundamental limitations of the approach


2  Questionable Evaluation Metrics:
The focus on distinguishing machine vs. human-generated text may be misguided - what matters more is the correctness and quality of reasoning
Binary classification between human/machine generation doesn't capture important nuances in response quality
The evaluation doesn't specifically assess improvement in reasoning, factual accuracy, or task-specific performance


3  Limited Experimental Scope:
Only tested on one base model (Llama-3-8B)
Uses a relatively small seed dataset (4,000 samples from UltraChat200k)
Lacks evaluation on diverse task types or specialized domains

4  Missing references:
There are other papers talking about the importance/pitfalls of discrimination for LLMs:
https://arxiv.org/abs/2404.04298
https://arxiv.org/abs/2311.08516
https://arxiv.org/abs/2310.01798

**Questions:**

See Weaknesses

---

### Official Review · Reviewer_3HEu · 2024-11-03

**Soundness:** 1
**Presentation:** 3
**Contribution:** 2
**Rating:** 3
**Confidence:** 4

**Summary:**

The authors are interested in iteratively fine-tuning a language model using high quality data generated by itself. Their proposed method (CoEvol)  improves both the model’s ability to generate language, but also the ability to discriminate between good and bad outputs (critique), such that good outputs are re-used for another iteration of training.

**Strengths:**

The authors demonstrate some empirically good results (Figure 4, 5, 6).

**Weaknesses:**

**[Lack of ablation]** I think a big missing piece of information is whether the improved generations are due to 1) training both the generator and critic model jointly, 2) due to better training data, or 3) both. Ie, does the improvement in language generation come from being a better discriminator somehow? Or, does the improvement come just from seeing more (presumably good) training data? This seems like an easy experiment to check, as one can just keep 2 separate models, in which one is being updated as a critic and the other is updated as the generator.

Ablating the above is important because numerous authors have already demonstrated that iterative training on a LM’s own output (with some filtering in between) is useful, many of which the authors cite. The authors claim that their novelty comes in jointly updating the model to be a better evaluator – but it is unclear if the improvements are because of jointly training or just from seeing better data.

**[Lack of comparison vs. prior work]** Speaking of prior work, the last point above is also unclear because the authors do not use any prior work as baselines in assessing generation quality. For instance, what is the difference in output quality when keeping the same (frozen) evaluator model vs. iteratively improving the evaluator?

**[Questionable human study]** Section 4.3.1+Figure 7: A single annotator evaluating on a sample size of 20 is not really a scientific result. I would believe that CoEvol leads to better generations according to human judges, but honestly this result is cringeworthy and rather hurts the overall quality of the paper. It is also unclear if the annotator is one of the co-authors. In Section 4.2, the text reads “we conduct a similar assessment with human evaluators *(plural)*”, which is misleading.

The same can be said about Section 4.2: For crafting the test set – can you elaborate on “additional prompts crowdsourced from the authors”? How much of the entire dataset consists of these additional prompts, what was the setup for crowdsourcing (who were the crowdworkers, what were the instructions given to them (ie, were they asked to generate entirely new prompts? Paraphrase existing ones? etc., Were there any steps taken to validate the quality of crowdworkers + their results?)). I hope it is not the case that the authors wrote the test cases.

**[Lack of comparison vs. prior work Part II]** Section 4.4: the argument is confusing, perhaps even self-contradicting? The authors first claim that the LM’s “self-generated outputs may become more similar to the high-quality seed data (human-generated data)”. If that’s the case, isn’t it desirable for the discriminative critic model to achieve close to 50% accuracy? Ie, its generations are indistinguishable from human-generated data? Ironically, the authors then show (Figure 8) that their discriminative critic model achieves a classification score of ~70%, while their baseline model achieves the desirable ~50% accuracy. The authors only compare against said baseline model on the discriminative task, but not on the language generation task, which is the actual end goal. Without this comparison, it is unclear if the 50% accuracy from the baseline model is actually due to better generations from the baseline model.

If higher discriminating accuracy is desirable - what is the explanation for classification accuracy dropping after 1 iteration? The current explanation is again perhaps contradictory: “This may be due to the improvement in the model’s instruction-following ability, which makes it more challenging to distinguish between high-quality seed data and self-generated data” – again, this seems to indicate that the desirable result is to reach 50% discriminatory accuracy. Furthermore, doesn’t this drop in accuracy imply that indeed the critic model should only be trained once (as opposed to iteratively), which is what prior work does?

Lastly, I wonder if the 70% vs. 50% discriminating accuracy is just further evidence that LM evaluators recognize and prefer their own outputs [Panickssery et al. LLM Evaluators Recognize and Favor Their Own Generations]. It would be helpful to hear the author’s thoughts on this related work, which is currently missing.

**[Lack of novelty]** To be honest, the suggested approach is not very interesting, though I did not take this point into consideration when assigning my score.

I am happy to raise my score if the above concerns are addressed.

**Questions:**

Prior work (Briesch et al. Large Language Models Suffer From Their Own Output: An Analysis of the Self-Consuming Training Loop) shows that such self-iterating methods has its pitfalls, such as a decrease in diversity. Can you comment on how your work relates to such findings?

The hyperparameter \alpha used for filtering seems like an imporant hyperparameter. How was it chosen?

---

### Note · Authors · 2024-12-05

I have read and agree with the venue's withdrawal policy on behalf of myself and my co-authors.